# Multidimensional Functional Profiling of Human Neuropathogenic *FOXG1* Alleles in Primary Cultures of Murine Pallial Precursors

**DOI:** 10.3390/ijms23031343

**Published:** 2022-01-25

**Authors:** Simone Frisari, Manuela Santo, Ali Hosseini, Matteo Manzati, Michele Giugliano, Antonello Mallamaci

**Affiliations:** 1Cerebral Cortex Development Laboratory, Department of Neuroscience, SISSA, Via Bonomea 265, 34136 Trieste, Italy; sfrisari@sissa.it (S.F.); msanto@sissa.it (M.S.); 2Neuronal Dynamics Laboratory, Department of Neuroscience, SISSA, Via Bonomea 265, 34136 Trieste, Italy; ahossein@sissa.it (A.H.); mmanzati@sissa.it (M.M.); michele.giugliano@sissa.it (M.G.)

**Keywords:** *FOXG1* syndrome, functional gene profiling, multidimensional gene profiling, neural fate choice, neural progenitor proliferation, neuron morphology, neuron activity, precision therapy

## Abstract

*FOXG1* is an ancient transcription factor gene mastering telencephalic development. A number of distinct structural *FOXG1* mutations lead to the “*FOXG1* syndrome”, a complex and heterogeneous neuropathological entity, for which no cure is presently available. Reconstruction of primary neurodevelopmental/physiological anomalies evoked by these mutations is an obvious pre-requisite for future, precision therapy of such syndrome. Here, as a proof-of-principle, we functionally scored three *FOXG1* neuropathogenic alleles, *FOXG1^G224S^*, *FOXG1^W308X^*, and *FOXG1^N232S^*, against their healthy counterpart. Specifically, we delivered transgenes encoding for them to dedicated preparations of murine pallial precursors and quantified their impact on selected neurodevelopmental and physiological processes mastered by Foxg1: pallial stem cell fate choice, proliferation of neural committed progenitors, neuronal architecture, neuronal activity, and their molecular correlates. Briefly, we found that *FOXG1^G224S^* and *FOXG1^W308X^* generally performed as a gain- and a loss-of-function-allele, respectively, while *FOXG1^N232S^* acted as a mild loss-of-function-allele or phenocopied *FOXG1^WT^*. These results provide valuable hints about processes misregulated in patients heterozygous for these mutations, to be re-addressed more stringently in patient iPSC-derivative neuro-organoids. Moreover, they suggest that murine pallial cultures may be employed for fast multidimensional profiling of novel, human neuropathogenic *FOXG1* alleles, namely a step propedeutic to timely delivery of therapeutic precision treatments.

## 1. Introduction

*Foxg1* encodes for an ancient transcription factor exerting a pleiotropic control on brain development. It primes pan-telencephalic [1], subpallial [2], and paleo-neo-pallial [3] programs. It stimulates neural precursors self-renewal [4] and modulates neural cell fate choice [5,6]. It biases neocortical neurons to specific layer identities [7,8,9,10] and promotes their morphological maturation [5,11,12,13]. Moreover, it sustains activity and excitability of these neurons [13,14], being in turn upregulated by neuronal activity [14,15]. As a consequence of that, experimental *Foxg1* manipulation in mouse models results in prominent, cognitive and behavioral, phenotypes. Loss of *Foxg1* leads to defective social interaction and impaired spatial learning and memory [12,16]. A concomitant misregulation of *Foxg1* in postnatal, gabaergic and glutamatergic neurons, is necessary and sufficient to trigger autism spectrum disorder-like (ASD-like) phenotypes [17].

In humans, a number of distinct *FOXG1* copy number variations and structural mutations have been described. They lead to an array of rare neuropathological scenarios, collectively referred to as the *FOXG1* syndrome [18]. Hemizygosity for *FOXG1* results in a variant of the classical Rett syndrome, with microcephaly, myelination anomalies, ASD symptoms (poor speech, stereotypies, sleep disorders, and social deficits) and epilepsy. *FOXG1* duplication may cause the West syndrome, including infantile spasms, aberrant EEG with hypsarrhythmia, epilepsy and severe cognitive deficits. Finally, a number of distinct structural *FOXG1* mutations, missense, nonsense and frameshift (>50 and >400 reported in SFARI [19] and ClinVar [20] databases, respectively), give rise to neurological outcomes from benign to very severe. These outcomes include subsets of symptoms peculiar to gene deletions and duplications. In particular, mutations falling upstream of the DNA-binding domain (DBD) (prevalently nonsense and frameshift), give rise to predominantly severe phenotypes, mutations within the DBD (often missense) have phenotypic correlates of diverse severity, more sporadic mutations located in the protein COOH-terminal half (including those leading to loss of Groucho-binding domain (GBD) and/or Jarid1B-binding domain (JBD)), can also result into variable neurological phenotypes.

It is reasonable to think that distinct FOXG1 protein domains (including those interacting with DNA, Groucho and Jarid1B), may be differentially needed for normal molecular control exerted by FOXG1 on various scenarios mastered by its gene. For this reason, *FOXG1* mutations could impact quite diversely the progression of distinctive neurodevelopmental subroutines mastered by the gene. Therefore, as a key step to rationally define (1) the type of precision intervention, suitable for prophylactic/therapeutic treatment of *FOXG1* mutations, and (2) the temporal window of opportunity for its delivery, a systematic functional dissection of the impact of these mutations on the main processes mastered by the gene is mandatory. Unfortunately, prolonged duration of human brain histogenesis and maturation, in vivo as well as in neuro-organoids, would make such dissection in homotypic neural tissue particularly time-demanding, so precluding its prompt therapeutic exploitation in case of *novel* mutations. The use of primary cultures of rodent neural tissue as a substrate for such dissection might offer a valid alternative. In fact, neurodevelopmental subroutines underlying rodent brain morphogenesis take place in much shorter times compared to primates (<1/10). Moreover, albeit very different, rodents and primates largely share fundamentals of brain development, including several key aspects of Foxg1/FOXG1 control of neocortical histogenesis [6,21,22,23,24], and Santo et al., unpublished results]. In this way, rodent neural cells can be an appealing substrate for fast, primary comparative characterization of human *FOXG1* alleles. 

Here, to assess the feasibility of this approach, we transduced a few, selected human neuropathogenic *FOXG1* alleles into murine neural cultures and systematically evaluated their impact on neurodevelopmental subroutines mastered by this gene, ranking their activities against the healthy *FOXG1* allele and a negative control (Figure 1). We focused on three mutations which likely affect the capability of the resulting protein to interact with chromatin and two key cofactors of it. For these mutations, heterozygous patient fibroblasts are available at public repositories, suitably for future, follow-up studies in human neural organoids. Specifically, we interrogated two alleles harboring missense mutations within the DNA-binding domain (DBD) (c.670G > A, corresponding to p.Gly224Ser [18,25], hereafter referred to as *FOXG1^G224S^*; and c.695A > G, corresponding to p.Asn232Ser [26,27], hereafter referred to as *FOXG1^N232S^*), as well as a third allele, encoding for a protein truncated upstream of GBD and JBD (c.924G > A, corresponding to p.Trp308Ter [28,29], hereafter referred to as *FOXG1^W308X^*). Briefly, we found that, in almost all scenarios investigated, *FOXG1^G224S^* consistently acted as a gain-of-function (GOF) allele and *FOXG1^W308X^* as a loss-of-function (LOF) one. *FOXG1^N232S^* performed like *FOXG1^WT^* or slightly weaker than it.

## 2. Results

### 2.1. Functional Characterization of FOXG1^G224S^, FOXG1^W308X^ and FOXG1^N232S^ along the Neuronogenic Lineage

It is known that mouse *Foxg1* inhibits cell cycle exit of neuronogenic neocortical precursors [4]. We found that this applies also to its human ortholog, *FOXG1* (Figure 2 and Appendix A). In fact, overexpressed in early murine pallial precursors by lentiviral Tet^ON^ transgenesis, *FOXG1* lowered the fraction of them activating the early postmitotic neuronal marker Tubb3, by 30.2 ± 3.2%, compared to *Placental alkaline phosphatase* (*Plap*)-expressing controls (*p* < 9.8 × 10^−5^, and *n* = 7,7). Remarkably, in the same experimental context, both *FOXG1^G224S^* and *FOXG1^W308X^* mimicked *FOXG1^WT^,* however outperforming it and performing weaker than it, respectively. Specifically, upon overexpression of these mutant alleles, Tubb3^+^ cell frequency declined by 65.3 ± 1.8% (*p* < 1.2 × 10^−5^ and *n* = 3,7) and 19.9 ± 4.5% (*p* < 0.019 and *n* = 3,7), respectively, compared to *Plap* samples. *FOXG1^N232S^* was conversely ineffective. 

Shut down to allow neuronogenic precursors exit from cell cycle, *Foxg1* is normally reactivated in newborn murine neurons [8], where it is needed to finely tune layer identity [9,10], dendritic architecture [5,11,12] as well as activity and excitability [14]. All that offers further valuable opportunities to functionally score mutant *FOXG1* alleles against their wild type counterparts.

In this respect, first, we investigated the impact of human *FOXG1* alleles on pallial neuron architecture. We found that, similarly to its murine paralog [11], *FOXG1^WT^* stimulated elongation and arborization of dendrites (Figure 3 and Appendix A). In fact, overexpressed in nascent neocortical neurons by lentiviral Tet^ON^ transgenesis under 100 ng/mL doxycycline (Figure 3A,B), it increased the average number of nodes connecting dendrite exit-points and end-points, *anN* (aligned nodes number), by 64.1 ± 12.7%, (with *p* < 10^−4^ and *n* = 18,26), and reduced the average internodal distance, *l* (an index anticorrelated to the dendritic arborization drive), by 17.4 ± 3.8 (*p* < 0.030 and *n* = 18,26). Then, we moved to mutant *FOXG1* alleles. Suspecting a likely gain-of-function associated to *FOXG1^G224S^* (as suggested by Figure 2 data), we tried to catch differences with its wild-type counterpart, comparing the morphometric performances of *FOXG1^G224S^* with those of *FOXG1^WT^* and *Plap* under reduced doxycycline levels (70 ng/mL, see Figure 3A,B). In these conditions, the behavior of *FOXG1^WT^*-overexpressing samples did not differ from *Plap* controls. Conversely, as expected, *FOXG1^G224S^* overexpression upregulated *anN*, by 18.1 ± 8.5% (*p* < 0.055 and *n* = 28,27), and reduced *l*, by 21.7 ± 5.0% (*p* < 0.053 and *n* = 28,27). Finally, as a putative LOF allele (see Figure 2 data), *FOXG1^W308X^* was profiled under 100 ng/mL doxycycline (Figure 3A,C). In these conditions, compared to *FOXG1^WT^*, it elicited only a barely appreciable increase of *anN* over *Plap* controls, namely 23.1 ± 11.5% in *FOXG1^W308X^* samples vs. 64.1 ± 12.7% peculiar to *FOXG1^WT^* ones (p_(*W308X/WT)*_ < 0.030 and *n* = 23,18), as well as no decline of *l*. 

Next, we investigated the impact of different *FOXG1* alleles on spontaneous electrical activity of neocortical neurons, by means of two dedicated assays (Figure 4).

In the first case, 3:1 mixes of E16.5 neocortical and archicortical neurons, engineered to express different *FOXG1* alleles and a GCaMP6s Ca^2+^ sensor [30], were allowed to age for 10 days and finally, optically sampled at 4 Hz for single neuron, intracellular Ca^2+^ fluctuations (Figure 4A). By this approach, we found that, compared to *Plap*-expressing controls, *FOXG1^G224S^* and *FOXG1^W308X^* increased the prevalence of active neurons, by 69.6 ± 2.9% (*p* < 0.001, *n* = 3,3) and 36.4 ± 5.9% (*p* < 0.013, *n* = 3,3), respectively (the corresponding increase displayed by *FOXG1^WT^* samples did not reach statistical significance). Next, we found that, compared to *Plap* controls, *FOXG1^WT^* upregulated the median frequency of Ca^2+^ events (likely corresponding to clusters of action potentials) by 87.8% (*p* < 10^−6^ and *n* = 16,18), and shifted the cumulative distribution of inter-event intervals (IEIs), reducing the median IEI by 42.9% (*p* < 10^−6^, and *n* = 1455,965). *FOXG1^G224S^* and *FOXG1^W308X^* upregulated the median frequency of Ca^2+^ events too, by 116.7% (*p* < 10^−6^, *n* = 17,18) and 52.1% (*p* < 1.6 × 10^−4^, *n* = 22,18), respectively, and shifted the cumulative distribution of inter-event intervals (IEIs) as well, decreasing the median IEI by 50.0% (*p* < 10^−6^, *n* = 1722,965) and 35.7% (*p* < 10^−6^, *n* = 2751,965), respectively. In synthesis, (1) similarly to its murine paralog [14], human *FOXG1^WT^* made spontaneous bursts of neuronal activity more frequent and, (2) in such context, *FOXG1^G224S^* and *FOXG1^W308X^* mainly acted as GOF and LOF variants of *FOXG1^WT^*, respectively.

As for the second assay, E16.5 neocortical neurons were again transduced at DIV1 by Tet^ON^-controlled, lentiviral transgenes, encoding for different FOXG1 alleles, and then allowed to age for a longer time. At DIV12, the transgenes were activated by doxycycline and, finally, at DIV21, the engineered cultures were sampled by multi-electrode array (MEA) analysis (Figure 4B). Interestingly, regardless of transgene identity, all (or almost all) electrodes were active (not shown), suggesting that all cultures were healthy and vital. Temporal articulation of their activity depended heavily on the transgene they expressed. Compared to *Plap* controls, FOXG1^WT^-overexpressing preparations displayed shorter bursts of electrical activity, with median burst duration (BD) and median number of spikes per burst (S/B) diminished by 78.02% and 53.83%, respectively (*p* < 10^−6^ and *n* = 725,660). The same cultures also showed a selective shortening of longer inter-burst intervals (IBIs), with the 75th-percentile IBI reduced by 14.61% (*n* = 723,657 and *p* < 10^−6^). Similar, however far more pronounced shifts were detectable in *FOXG1^G224S^*-GOF preparations, where median BD, median S/B and 75th-percentile IBI were decreased by 97.01%, 92.44% and 49.83%, respectively (all with *p* < 10^−6^, and *n* = 1758,660 and 1755,657). Conversely, the *FOXG1^W308X^* transgene diminished median BD and S/B only by 17.77% and 38.60%, respectively (both with *p* < 10^−6^ and *n* = 500,660) and increased 75th-percentile IBI by 107.97% (with *p* < 10^−6^ and *n* = 497,597). In synthesis, (1) compared to Plap controls, hyper-active, *FOXG1^WT^*-overexpressing cultures took less time to trigger burst-suppressing responses as well as to initiate new bursts, and (2) *FOXG1^G224S^* and *FOXG1^W308X^* mainly acted as GOF and LOF variants of *FOXG1^WT^*, respectively.

The phenotypes outlined above, specifically elicited by mutant *FOXG1* alleles within the neuronogenic lineage, likely originated from an altered transcriptional control of molecular mediators of the corresponding neurodevelopmental subroutines. To corroborate this inference and unveil additional, valuable “sensors” of *FOXG1* alleles’ performances, we selected two mini-panels of genes, which are specifically down- and up-regulated in engineered *Foxg1*-GOF neocortical neuron cultures [31], and we systematically scored their mRNA levels upon *FOXG1^G224S^* and *FOXG1^W308X^* activation, against *FOXG1^WT^*-overexpressing samples (Figure 5). A subset of these genes, *^endo^Foxg1*, *Gad1*, *Gad2*, *Gabra1*, and *Grin2a*, was downregulated by human *FOXG1* similarly to murine *Foxg1*. Remarkably, their down-regulation was exacerbated upon *FOXG1^G224S^* over-expression and attenuated following *FOXG1^W308X^* over-expression. Other genes, *Arc, Hes1, Npas4, Grik3, Grik4, Cacna2d2, Scn11a* and *Grin2c*, were conversely up-regulated by human FOXG1 similarly to murine Foxg1. Even here, gene responses to *FOXG1^G224S^* and *FOXG1^W308X^* were exacerbated and attenuated, respectively, compared to *FOXG1^WT^*, with two exceptions: referring to *FOXG1^WT^*, the *Scn11a* expression gain elicited by *FOXG1^G224S^* was less than halved (*p* < 10^−5^, *n* = 4,4), and *Grin2c* levels peculiar to *FOXG1^W308X^* samples largely overcame *FOXG1^WT^* samples (*p* < 0.001, *n* = 4,4), equaling *FOXG1^G224S^* ones. Altogether, these data suggest that *FOXG1^G224S^* and *FOXG1^W308X^* generally impact regulation of neuronal *FOXG1*-sensitive genes as GOF and LOF variants of *FOXG1^WT^*.

### 2.2. Impact of FOXG1^G224S^, FOXG1^W308X^ and FOXG1^N232S^ Alleles on Astrogenesis Progression

It is known that mouse *Foxg1* counteracts NSCs progression to the astrogenic lineage [6]. We found that this applies also to human *FOXG1.* In fact, overexpressed in early pallial NSCs and compared to *Plap* controls, *FOXG1* lowered the frequency of differentiated astroglial clones which originated from these cells in our standard differentiative conditions (3d without Lif supplementation), from 12.7 ± 2.0% to 8.2 ± 1.4% (*p* < 0.056, *n* = 4,4) (Figure 6A,B and Appendix A (top)). Biasing cultures towards astrogenesis (by means of a longer differentiation time, 4d vs. 3d, and terminal Lif stimulation), performances of *FOXG1^WT^*-overexpressing NSCs became indistinguishable from *Plap* controls, however, in these new experimental conditions, *FOXG1^G224S^* was still able to decrease astroglial clones, from to 18.8 ± 3.8% to 10.0 ± 2.0% (*p* = 0.055, *n* = 3,3) and increase mixed ones, from 44.2 ± 2.2% to 56.6 ± 0.9% (*p* < 0.003, *n* = 3,3) (Figure 6A,C and Appendix A (top)). Next, back to standard differentiative conditions (allowing the emergence of anti-astrogenic *FOXG1^WT^* activity), *FOXG1^W308X^* did not affect the frequency of astroglial clones, while conversely reducing neuronal ones, from 42.7 ± 3.5% (peculiar to Plap-controls) to 34.1 ± 1.4% (*p* < 0.032, *n* = 4,3) (Figure 6A,D and Appendix A (top)). Finally, upon 4d differentiation and terminal Lif stimulation, compared to *Plap* controls, *FOXG1^N232S^* elicited an increase of neuronal clones, 54.6 ± 2.7% vs. 44.0 ± 4.9% (*p* < 0.052, *n* = 4,4), and a reduction of mixed ones, 26.8 ± 3.2%, vs. 35.6 ± 2.1% (*p* < 0.031, *n* = 4,4), substantially phenocopying *FOXG1^WT^* (Figure 6A,E and Appendix A (top)). In synthesis, concerning their impact on NSC astroglial fate choice, *FOXG1^WT^* antagonized astrogenic progression like its murine ortholog, *FOXG1^G224S^* worked as a GOF allele, *FOXG1^W308X^* acted as a LOF and/or a dominant negative (DN) allele, and *FOXG1^N232S^* performed similarly to its healthy counterpart.

We had preliminary evidence that, while inhibiting NSC progression to astrocyte-committed progenitors, mouse *Foxg1* promotes self-renewal of these progenitors (Santo et al. unpublished data). We tried to replicate this result with *FOXG1^WT^*. Then, we scored the impact of *FOXG1^G224S^*, *FOXG1^W308X^* and *FOXG1^N232S^* alleles on the proliferating fraction of astrocyte progenitors, against their healthy counterpart and *Plap* controls. For these purposes, Foxg1/FOXG1-encoding transgenes were delivered by lentiviral vectors to E12.5+DIV10 neural cultures, highly enriched in astrocyte progenitors and, seven days later, such cultures were profiled by Ki67/Gfap-immunofluorescence. It turned out that upon human *FOXG1^WT^* overexpression, the Ki67^+^Gfap^+^/Gfap^+^ astroblast proliferating fraction was increased by 2.73 ± 0.09-folds compared to *Plap* controls (*p* < 1.45 × 10^−10^, *n* = 11,10), similarly to murine *Foxg1* overexpression (3.18 ± 0.32-folds, *p* < 3.45 × 10^−6^, *n* = 3,10). Again, compared to *Plap* controls, the same fraction was also upregulated by *FOXG1^G224S^*, *FOXG1^W308X^* and *FOXG1^N232S^*, by 4.88 ± 0.20, 1.27 ± 0.15 and 2.42 ± 0.16-folds, respectively (with *p* < 3.20 × 10^−9^, *p* < 0.086 and *p* < 1.34 × 10^−5^, respectively, and *n* = 3,8,4,10). Interestingly, the resulting differences in proliferation gains displayed by *FOXG1^G224S^*, *FOXG1^W308X^* and *FOXG1^N232S^* alleles with respect to *FOXG1^WT^* were all statistically significant (*p* < 5.65 × 10^−8^, *p* < 2.80 × 10^−8^, and *p* < 0.049, respectively) (Figure 6F,G and Appendix A (bottom)). In a few words, similarly to their impact on astroblasts proliferation, *FOXG1^G224S^*, *FOXG1^W308X^* and *FOXG1^N232S^* worked as strong-GOF-, strong-LOF and mild-LOF variants of *FOXG1^WT^*.

Finally, to tentatively generalize the scores achieved by mutant alleles in previously described NSC fate choice and astroblast proliferation assays and unveil additional “molecular sensors” of *FOXG1* alleles’ performances in the astroglial lineage, we selected two small gene sets displaying opposite expression trends, downwards (*^endo^Foxg1*, *Kcnk2*, *Glt1* and *Cnx43*) and upwards (*Kir4.1*, *Adk* and *mGlur5*), upon *Foxg1* overexpression in primary cultures of murine astrocytes (Santo et al. unpublished results), and we systematically evaluated the corresponding mRNA levels upon *FOXG1^G224S^* and *FOXG1^W308X^* activation, against *FOXG1^WT^*-overexpressing samples. Unexpectedly, *FOXG1^G224S^* outperformed *FOXG1^WT^* only in cases of *^endo^Foxg1* (normalized against *Plap*-samples, 0.42 ± 0.05 vs. 0.62 ± 0.03, respectively, with *p* < 0.008, *n* = 3,4), *Kcnk2* (0.57 ± 0.10 vs. 0.79 ± 0.08, with *p* < 0.073 and *n* = 4,4) and *Kir4.1* (1.45 ± 0.04 vs. 1.12 ± 0.06, with *p* < 0.002 and *n* = 4,4). Similarly, *FOXG1^W308X^* performed weaker than *FOXG1^WT^* only in cases of *^endo^Foxg1* (0.97 ± 0.12 vs. 0.62 ± 0.03, respectively, with *p* < 0.013 and *n* = 4,4), *Cxn43* (0.74 ± 0.04 vs. 0.63 ± 0.04, with *p* < 0.05 and *n* = 4,4), and *Adk* (0.99 ± 0.04 vs. 1.23 ± 0.02, with *p* < 0.001 and *n* = 4,4). Other responses elicited by mutant alleles did not differ from those triggered by *FOXG1^WT^* or ranked along a different progression (Figure 7). These results suggest that, compared to their healthy counterpart, specific mutant *FOXG1* alleles might impact mature astrocyte functions according to poorly predictable trends, different from the relatively simple ones emerging from inspection of earlier neurodevelopmental subroutines. More in general, expression dynamics of small gene sets evoked by neuropathogenic mutations may hardly suffice to predict macroscopic, histogenetic consequences of such mutations.

### 2.3. Functional Characterization of FOXG1^G224S^, FOXG1^W308X^ and FOXG1^N232S^ along the Oligodendrogenic Lineage

It was previously reported that mouse *Foxg1* overexpression in pallial NSCs reduces their oligodendroglial output [5]. Here, first we verified that this also applies to human *FOXG1*. For this purpose, we delivered a transgene encoding for it to primary cultures of early pallial precursors and allowed these precursors to age in vitro past the neuronogenic window, while keeping the transgene specifically on in NSCs. Next, we moved cells to a pro-differentiative medium. Finally, we evaluated the frequency of their descendants activating the pan-oligodendroglial CNPase marker. As expected, *FOXG1^WT^* reduced such frequency, from 26.1 ± 3.7% (peculiar to *Plap* controls) to 13.1 ± 2.6% (*p* < 0.015, *n* = 4,3). Moreover, *FOXG1^WT^* also lowered the CNPase^+^/Gfap^+^ cell ratio, from 0.65 ± 0.02 to 0.33 ± 0.06 (*p* < 0.03, *n* = 4,3), suggesting a further oligodendrogenic-to-astrogenic shift of the engineered culture. Then, we inspected sister cultures overexpressing *FOXG1^G224S^* and *FOXG1^W308X^*. Unexpectedly, both alleles quantitatively phenocopied their healthy counterpart, suggesting that the corresponding mutations may not differentially perturb the oligodendrogliogenic bias of pallial NSCs (Figure 8A,B and Appendix A (top)).

It was also previously reported that *Foxg1* promotes proliferation of oligodendroglial progenitors, delaying their terminal differentiation [32]. Again, we verified that this also applies to human *FOXG1*. To this aim, first, we allowed early pallial precursors to age in proliferative medium until the in vitro equivalent of the peak astrogenic time (P4). Next, we transferred such precursors to a T3/IGF1-supplemented, pro-differentiative medium and we transduced them with a pPgk1/Tet^ON^ controlled *FOXG1^WT^* transgene. One day later, we activated the transgene, supplementing the medium by doxycycline. Finally, three days more later, we evaluated the proliferating, Ki67^+^ fraction of CNPase^+^ oligodendroglial derivatives. As expected, compared to *Plap*-transduced controls, *FOXG1^WT^* increased such fraction by 83.9 ± 4.5% (*p* < 0.001, *n* = 3,3). Then, we scored sister cultures overexpressing *FOXG1^G224S^*, *FOXG1^W308X^* and *FOXG1^N232S^*. All three mutant alleles increased the oligodendroglial proliferating fraction, by 150.7 ± 10.1%, 53.5 ± 13.8% and 73.7 ± 17.0%, respectively (with p_vs-Plap_ < 10^−4^, p_vs-Plap_ < 0.017, p_vs-Plap_ < 0.010, respectively, and *n* = 3,3,3,3). Differences among mutant *FOXG1* alleles and *FOXG1^WT^* were statistically significant in case of *FOXG1^G224S^* and *FOXG1^W308X^* (with *p* < 0.001 and *p* < 0.050, respectively). All that suggests that pro-proliferative activities displayed by the different alleles within the oligodendrogliogenic lineage differ, according to the declining, *FOXG1^G224S^ > FOXG1^WT^* = *FOXG1^N232S^* > *FOXG1^W308X^* progression (Figure 8C,D and Appendix A (bottom)).

## 3. Discussion

In this study, we systematically compared the functional performances of three human *FOXG1* neuropathogenic alleles, with their healthy counterpart. These alleles were: *FOXG1^G224S^* (harbouring a missense mutation within the DNA-binding domain), *FOXG1^W308X^* (encoding for a truncated protein missing both Groucho- and Jarid1B-binding domains), and *FOXG1^N232S^* (harbouring another missense mutation within the DNA-binding domain). To characterize these alleles, we delivered Tet^ON^-controlled transgenes encoding for them to dedicated preparations of murine pallial precursors and quantified their impact on a number of neurodevelopmental and physiological processes mastered by *Foxg1*. We monitored the neuronogenic axis, scoring NSC commitment to neuronogenesis, neuron generation rates, dendrites elongation and arborization, neuronal activity, and expression levels of a representative set of neuronal genes (Figure 6, Figure 2, Figure 3, Figure 4 and Figure 5, respectively). We investigated the astrogenic axis, evaluating NSC commitment to astrogenesis, proliferative bias of astrogenic progenitors, and expression levels of a mini-set of astroglial genes (Figure 6 and Figure 7, respectively). To a lesser extent, we also focused on the oligodendrogenic axis, scoring NSC progression to oligodendrogenesis and proliferative bias of oligogenic progenitors (Figure 8). We found that, in all three axes, the human *FOXG1^WT^* allele acted similarly to its murine ortholog. Moreover, we discovered that, in most cases, *FOXG1^G224S^* and *FOXG1^W308X^* performed as a GOF and a LOF variants of *FOXG1^WT^*, respectively. This did not apply to NSC progression to oligodendrogenesis, counteracted with comparable strength by all *FOXG1* alleles subject of investigation. Finally, in a limited number of cases, *FOXG1^N232S^* acted as a mild LOF variant of *FOXG1^WT^* or phenocopied it (for a synopsis, see Table 1). 

Remarkably, our study was performed in wild-type neocortical precursors, engineered to overexpress exogenous *Foxg1* alleles and, therefore, characterized by cumulative *Foxg1*-mRNA levels well above the physiological baseline (References [11,14,33] and Appendix A). Notwithstanding that, we are confident that *Foxg1* biological activities detected by our experimental platform are genuine, as they nicely mirror the results of a number of *loss-of-function* studies performed on the same gene by other Teams and ours. This applies to investigation of NSC fate choice [5,6], proliferation of neuronogenic [4] and oligodendrogenic [32] progenitors, dendritic architecture [11] and neuronal activity [14].

The consistent, gain- or loss-of-function outcome emerging from *FOXG1^G224S^* and *FOXG1^W308X^* overexpression in distinctive neurodevelopmental scenarios is remarkable. It suggests that, beyond our original expectations, shared core molecular mechanisms mediate FOXG1 impact on different gene-sets active in such scenarios. Remarkably, that might ease precision therapeutic tuning of *FOXG1* activity in patients heterozygous for these alleles.

Among results of this study, of particular interest is the impact that *FOXG1^G224S^* and *FOXG1^W308X^* exert on neocortical neuronal activity, namely a feature exquisitely sensitive to *Foxg1* expression levels [14]. Reminiscent of similar phenomena co-occurring in neocortical cultures upon pharmacological modulation of the gabaergic-vs-glutamatergic balance [34], the shifts displayed by cumulative IBI and BD distributions in *FOXG1^G224S^* and *FOXG1^W308X^* engineered DIV21 cultures are consistent with the corresponding changes in IEIs distribution detectable in DIV10 cultures. As such, they suggest that electrical aberrancies evoked by neuropathogenic *FOXG1* alleles are not contingent to a restricted time window but can affect nerve cells at different stages of their development and maturation.

Intriguingly, the only histogenetic test where *FOXG1^G224S^* and *FOXG1^W308X^* quantitatively phenocopied *FOXG1^WT^* was the NSC-to-oligodendrocyte progression assay (Figure 8A). Rather than a selective insensitivity of such progression to G224S and W308X mutations, this could alternatively reflect the progressive decline of *Foxg1* expression in aging, pallial stem cells [6] and the likely need of extremely low *Foxg1* levels to allow late progression of these cells to oligodendrogenesis. If so, even NSCs transduction with *FOXG1^W308X^* (providing “weak” Foxg1-activity) might be sufficient to achieve maximal, *Foxg1*-dependent inhibition of NSC progression to oligodendrogenesis, making functional differentiation of *FOXG1* alleles problematic in this context. In this respect, the alternative implementation of this assay in pallial cells from *Foxg1^+/-^* mouse donors (maybe under lower doxycycline concentrations) could help “latent” differences among *FOXG1* alleles in oligogenesis control to emerge.

Last, despite substantial functional conservation emerging between mouse and human *Foxg1* orthologs, the murine gene promoted neuronogenic fate choice stronger than its human counterpart (Figure 6B) and, compared to it, exerted a more pronounced impact on the neuronal transcriptome (Figure 5). This probably reflects a better adaptation of Foxg1 proteins to their homologous cell environments.

Concerning mechanistic origin of gain- and loss-of-function phenotypes displayed by *FOXG1^G224S^* and *FOXG1^W308X^*, respectively, a few considerations are in order. In both neurons and astrocytes engineered to overexpress exogenous *FOXG1* alleles, endogenous-*Foxg1*-mRNA levels are down-regulated (Figure 5 and Figure 7), and total *Foxg1*/*FOXG1*-mRNA levels (evaluated via RTPCR amplification of a shared cds-fragment) are mainly representative of exogenous transgenes (Appendix A). That said, no significant differences were apparently detectable between *FOXG1^G224S^* and *FOXG1^WT^* expression levels, suggesting that, rather than reflecting differential abundance of its product, enhanced *FOXG1^G224S^* performances may be due to intrinsic structural differences among the pathogenic and the healthy protein. On the other side, *FOXG1^W308X^* expression levels hugely overcame *FOXG1^WT^* ones (Appendix A), indicating that that LOF performances of the former allele hardly originated from its defective expression and rather suggesting that they could have been due to dominant-negative effects exerted by FOXG1^W308X^ on healthy Foxg1 protein.

Even if we cannot rule out de novo neuropathogenic functions associated to mutant alleles, altogether, results of our study suggest that the neurological scenarios characterizing patients heterozygous for *FOXG1^G224S^, FOXG1^W308X^* and *FOXG1^N232S^* [18,26,27,28], may largely originate from a *quantitatively* misregulated control of *FOXG1*-dependent histogenetic processes. Of course, this inference needs to be corroborated by histogenetic profiling of neural organoids originating from patient-specific iPSCs, as well as by an unbiased molecular profiling of neural cells heterozygous for mutant *FOXG1* alleles. Should it be confirmed, this inference could pave the way to patients’ treatment via precision compensatory tuning of their *FOXG1*-mRNA levels, e.g., by means of RNA interference or RNA-activation [15], namely an approach potentially scalable to other neuropathogenic *FOXG1* mutations. In this respect, an obvious distinction should be posed between: (1) neurodevelopmental aberrancies occurring during the first half of gestation and (2) later pathogenic phenomena (e.g., altered maturation schedule of astroglia and oligodendroglia, abnormal shaping of dendritic trees and misregulation of neuron activity/excitability), taking place during later gestational life as well as after birth. Despite recent advancements in experimental prenatal diagnosis of fetal mutations [35,36], due to their extremely early occurrence, the former will be hardly tractable. Conversely, as for the latter, the availability of a standardized platform for fast multidimensional profiling of novel mutant alleles (such as the one described here), could pave the way to timely therapeutic interventions, eventually resulting in an appreciable mitigation of the corresponding neurological symptoms.

## 4. Conclusions

Main conclusions of this study are:(a)Primary murine neural cultures can be succesfully employed for fast, functional multidimensional profiling of novel, human neuropathogenic *FOXG1* alleles, propedeutically to *timely* delivery of appropriate precision therapies;(b)Among the three mutant *FOXG1* alleles subject of the present pilot study, *FOXG1^G224S^* (encoding for a protein with altered DBD) and *FOXG1^W308X^* (encoding for a protein missing GBD and JBD) generally performed as a GOF and a LOF allele, respectively, while *FOXG1^N232S^* (encoding for another protein with altered DBD) acted as a mild LOF allele or phenocopied *FOXG1^WT^*;(c)Although de novo neuropathogenic functions associated to such mutations cannot be ruled out, the concordant functional dissimilarities displayed by each mutant allele, when ranked against its healthy counterpart in a variety of neurodevelopmental and physiological contexts, suggest that the neuropathological traits peculiar to patients heterozygous for *FOXG1^G224S^*, *FOXG1^W308X^* and FOXG1N232S mainly stem from *abnormal quantitative tuning* of *FOXG1*-controlled processes.

## 5. Materials and Methods

### 5.1. Mice and Embryo Dissection

Animal handling and subsequent procedures were in accordance with European and Italian laws (European Parliament and Council Directive of 22 September 2010 [2010/63/EU]; Italian Government Decree of 4 March 2014, no. 26). Experimental protocols were approved by SISSA OpBA (Institutional SISSA Committee for Animal Care) and authorized by the Italian Ministery of Health (Auth. No. 22DAB.N.4GU). *Mtapt^EGFP/+^* [37] or wt CD1 males were mated to wt CD1 females (purchased from Envigo Laboratories, Italy) and maintained at the SISSA mouse facility. Embryos were staged by timed breeding and vaginal plug inspection. Pregnant females were sacrificed by cervical dislocation. Embryonic cortices and hippocampus were dissected out in cold 1X-PBS, supplemented with 0.6% glucose, under sterile conditions. *Mtapt^EGFP/+^* E12.5 embryos were distinguished from their wild type littermates by inspection under fluorescence microscopy.

### 5.2. Lentiviral Vectors Packaging and Titration

Third generation self-inactivating (SIN) lentiviral vectors (LVs) were generated and titrated as previously described [5].

Lentiviral vectors employed in this study include:

LV_pPgk1-rtTA2S-M2 [38];

LV_pNes-rtTA-M2, aka pNes/hsp68-rtTA2S-M2 [5];

LV_pTα1-rtTA2S-M2 [5];

LV_pSyn-rtTA2S-M2 [14];

LV_TREt-PLAP [6];

LV_TREt-mmuFoxg1^wt^ [39];

LV_TREt-hsaFOXG1^wt^ [built by transferring the AgeI-SalI fragment from pUC57 “h-F1WT” plasmid (Gene Universal) into the AgeI-SalI digested LV_TREt-IRES2-EGFP [40]];

LV_TREt-hsaFOXG1^G224S^ [built by transferring the AgeI-SalI fragment from pUC57 “h-F1-G224S” (Gene Universal) into the AgeI-SalI digested LV_TREt-IRES2-EGFP [40]];

LV_TREt-hsaFOXG1^W308X^ [built by transferring the AgeI-XhoI fragment from pUC57 “h-F1-W308X” (Gene Universal) into the AgeI-SalI digested LV_TREt-IRES2-EGFP [40]];

LV_TREt-hsaFOXG1^N232S^ [built by transferring the BsaBI-XmaI fragment from pUC57 “h-F1-N232S” (Gene Universal) into BsaBI-XmaI digested LV_TREt-hsaFOXG1^wt^];

LV_GCaMP6s [built by transferring the the BamHI/HindIII filled GCaMP6s-cds fragment from pAAV.Syn GCaMP6s.WPRE.SV40 (Addgene, #100843) into BamHI/XhoI filled LV_pSyn-rtTA^2S^-M2].

When not otherwise stated, each LV was employed at a multiplicity of infection (moi) of 8. Murine neural cells were transduced at densities of 750–1000 cells/μL. As previously described [5], and according to our experience (not shown), these conditions are sufficient to effectively co-transduce almost the totality of neural cells [33].

### 5.3. Primary Cortical Precursors Cultures

#### 5.3.1. Depending on the Assay, Cultures Were Set as Follows

Neuronogenic rate assays (Figure 2). E12.5 neocortical precursors were obtained dissecting neocortices from wt mice and dissociating them to single cells by gentle pipetting. Aliquots of 3 × 10^5^ cortical precursors were resuspended each in 400 μL of “pro-proliferative medium” and cultured in uncoated, 2 cm^2^ wells of 24 multiwell plates (DB Falcon). Neural cells were acutely infected by specific lentivector mixes (as in Figure 2) and allowed to grow as floating spheres. The dissection/infection day was referred to as “day in vitro 0” (DIV0). Tet^ON^-modulated transgenes were activated at DIV0 by 100 ng/mL doxycycline (Sigma #D9891-10G), and kept further on by doxycycline hemi-supplementation every 2 days. At DIV2.5, spheres were trypsinized and precursors were replated in pro-proliferative medium, at the initial density. At DIV5, spheres were again dissociated to single cells. Aliquots of 3 × 10^5^ cells were transferred to 0.2 mg/mL poly-L-lysine (Sigma #P2636)-pre-treated coverslips, and allowed to settle 1 h under “acute differentiative medium” (Dulbecco’s Modified Eagle Medium (Invitrogen) containing 10% fetal bovine serum (FBS), and penicillin/streptomycin). Cells were finally fixed by 4% PFA for subsequent immunofluorescence analysis. 

Dendrite morphometry assays (Figure 3). E12.5 neocortical precursors were obtained dissecting neocortices from *Mtapt^EGFP/+^* or wt mice and dissociating them to single cells by gentle pipetting. Cells were cultured in uncoated 24 multiwell plates (DB Falcon). Specifically, aliquots of 3 × 10^5^ cells were transferred to each well, at 1.5 × 10^5^/cm^2^, in 400 μL of “pro-proliferative medium” and allowed to grow as floating spheres. The dissection/infection day was referred to as “day in vitro 0” (DIV0). Moreover, cells were acutely infected by specific lentivector mixes (see details in Figure 3) and TetON-modulated transgenes were activated at DIV0 by 100 ng/mL doxycycline. At DIV2, spheres were dissociated to single cells, “green” *Mtapt^EGFP/+^* and “black” wt neurosphere derivatives were mixed at a 1:500 ratio (that was intended to ease morphological profiling of single neurons, albeit belonging to a dense ensemble), and aliquots of 3 × 10^5^ pre-mixed cells were transferred to 0.2 mg/mL poly-L-lysine (Sigma #P2636)-pre-coated, 2 cm^2^-coverslips, under 300 μL of “pro-differentiative medium”. Cells were cultured up to DIV12 and then blocked in 4% PFA, prior to morphometric analysis.

Neuron profiling by Ca^2+^ imaging and MEA analysis (Figure 4). Neocortical and hippocampal primordia dissected from E16.5 mouse brains were chopped separately to small pieces for 5 min, in the smallest volume of ice-cold 1X PBS—0.6% glucose—0.1% DNaseI solution. The minced tissues were then suspended and digested in 0.25 mg/mL trypsin- 4 mg/mL DNaseI for 5 min at 37 °C. Digestion was stopped by adding ≥1.5 volumes of Neurobasal A/10%FBS. Chunks of neural tissue were spun down and transferred to specific differentiative media (see below). The suspensions were pipetted 5–8 times with a P1000 and P200 Gilson pipettes and undissociated tissue was left to sediment for 2 min. The supernatants were harvested, and the living cells counted. 

In case of samples for Ca^2+^ imaging assays, aliquots of 2.5 × 10^5^ cells (3:1 premixed, neocortical and hippocampal ones) were plated on 2 cm^2^ coverslips (pretreated by 0.1 mg/mL poly-L-Lysine), in 400 µL of “pro-differentiative medium” (further supplemented by 8% FBS and 25 μM L-glutamate). Lentiviral mixes were added to the cultures the day after plating (DIV1). Tet^ON^ regulated transgenes were activated by 100 ng/mL doxycycline at DIV2 and kept on until the end of the experiment (DIV10) by doxycycline hemi-supplementation every 2 days.

In case of samples for MEA-analysis, aliquots of 1.8 × 10^6^ purely neocortical cells were plated on each MEA (pre-sonicated and pre-treated by 0.1% wt/vol polyethyleneimine, PEI, Sigma-Aldrich), at 6500 cells/mm^2^, in 1 mL of “MEA medium” [MEM (ThermoFischer), supplemented by 20 mM Glucose (ThermoFischer), 50 µg/mL Gentamycine (Gibco), 50µM L-glutamine (Gibco), and, limited to DIV0-DIV8, 10% Heat-Inactivated Horse Serum (Sigma-Aldrich)]. Evaporation was strongly reduced by sealing MEAs with PDMS caps. Every two days, 300 µL of old medium were removed and replaced with 350 µL of fresh medium. Lentiviral mixes were added to the cultures the day after plating (DIV1). Tet^ON^ regulated transgenes were activated by 100 ng/mL doxycycline at DIV12 and kept on until the end of the experiment (DIV21) by doxycycline hemi-supplementation every 2 days.

Neuronal mRNA profiling (Figure 5). Neocortical tissue from E16.5 mice was dissected and processed to single cells as described in Neuron profiling by Ca^2+^ imaging and MEA analysis. Cells were resuspended in “pro-differentiative medium”, and aliquots of 600,000 cells were plated on 0.1 mg/mL poly-L-Lysine pre-treated wells of 12-multiwell plates, in 600 µL of medium. Lentiviral mixes were added to the cultures the day after plating (DIV1). Tet^ON^ regulated transgenes were activated by 100 ng/mL doxycycline at DIV2 and kept on until the end of the experiment (DIV8) by doxycycline hemi-supplementation every 2 days. Finally, cultures were processed for RNA extraction, by Trizol^TM^.

Astrocyte fate choice assays (Figure 6A–E). Pallial tissue was dissected from E11.5 mouse embryos, mechanically dissociated to single cells by gentle pipetting and kept in “pro-proliferative medium”, at 1.5 × 10^5^ cells/cm^2^, in uncoated 24 multiwell plates, for 4 days. Cells were infected with dedicated LV-sets just after the dissection (DIV 0), and Tet^ON^-controlled transgenes were kept on by 2 μg/mL doxycycline throughout the culturing window. At DIV4, cells were trypsinized, resuspended in “pro-differentiative medium” (β-mercaptoethanol-depleted and further supplemented by 3% FBS, to final 5% FBS), and attached to poly-D-lysine (0.2 mg/mL, Sigma #A-003-E)-coated coverslips, at 16,800 cells/cm^2^. In case of Figure 6B,D assays, cultures were fixed at DIV7 in 4% PFA for analysis. In case of Figure 6C,E assays, cultures were supplemented by 30 ng/mL LIF (Sigma ESG1106) at DIV7, and fixed one more day later. 

Astrocyte’s proliferation assay (Figure 6F,G). Neoortices were dissected from E12.5 mouse embryos and mechanically dissociated to single cells by gentle pipetting. Neural cells were resuspended in “pro-proliferative medium”, at 6 × 10^5^ cells/mL, and cultured in T25 Flasks (Corning #430639), 3 × 10^6^ cell/flask. Floating neurospheres were trypsinized every 3.5 days and, at DIV 10, dissociated to single cells. Neural cells were resuspended in “pro-differentiative medium” (further supplemented by 8% FBS, to final 10% FBS), attached to 0.1 mg/mL poly-L-lysine-pretreated coverslips, at a density of approximately 1.5 × 10^5^ cells/cm^2^, and acutely transduced with a dedicated LV mix. Tet^ON^-controlled transgenes were activated the same day (DIV10) by 2 μg/mL of doxycycline and kept on by doxycycline hemi-supplementation every 2 days. One week later (at DIV17), cells were fixed in 4% PFA for analysis. 

Astrocytes mRNA profiling (Figure 7). Pallial primordia were dissected from E12.5 mouse embryos, mechanically dissociated to single cells by gentle pipetting and resuspended in “pro-proliferative medium”. Aliquots of 3 × 10^6^ cells were cultured in T25 Flasks, at 6 × 10^5^ cells/mL, and the resulting floating neurospheres were trypsinized every 3.5 days. At DIV 14, single cells originating from spheres dissociation were resuspended in “pro-differentiative medium” (further supplemented by 8% FBS, to final 10% FBS), attached to 0.1 mg/mL poly-L-lysine-treated coverslips, at a density of approximately 6 × 105 cells/cm2, and acutely transduced with a dedicated LV mix. Tet^ON^-controlled transgenes were activated by 500 ng/mL doxycycline at DIV18 and kept on by doxycycline hemi-supplementation every two days until the end of the procedure. Finally, at DIV25, cells were processed for RNA extraction by Trizol^TM^ Reagent (ThermoFisher). 

Oligodendrocyte differentiation assays (Figure 8A,B). Neocortices were dissected from E12.5 mouse embryos, mechanically dissociated to single cells by gentle pipetting. Cells were acutely engineered with dedicated LV mixes and kept in “pro-proliferative medium”, at 7.5 × 10^5^ cells/mL, in 24-multiwell plates, 3 × 10^5^ cells/well. Tet^ON^-controlled transgenes were kept on by 100 ng/mL of doxycycline throughout the experiment. Floating neurospheres were trypsinized every 3.5 days. At DIV 12, single cells originating from spheres dissociation were resuspended in “pro-differentiative medium”, attached to 0.1 mg/mL poly-L-lysine-treated coverslips, at a density of approximately 1.5 × 10^5^ cells/cm^2^. Seven days later, cells were fixed in 4% PFA for analysis. 

Oligodendrocyte progenitor proliferation assays (Figure 8C,D). Pallial primordia were dissected from E11.5 mouse embryos and mechanically dissociated to single cells by gentle pipetting. Neural cells were kept in “pro-proliferative medium”, at 6 × 10^5^ cells/mL, in T25 Flasks (Corning #430639), 3 × 10^6^ cells/flask. Floating neurospheres were trypsinized every 3.5 days and, at DIV 11, they were ultimately dissected to single cells. Cells were resuspended in “pro-differentiative medium” (further supplemented by 3% FBS, 30 ng/mL T3 (Sigma #T6397) and 50 ng/mL IGF (Sigma #I8779)), attached to 0.1 mg/mL poly-L-lysine-treated coverslips, at a density of approximately 1.5 × 10^5^ cells/cm^2^, and acutely transduced with a dedicated LV mix. Tet^ON^-controlled transgenes were activated at DIV12, by 100 ng/mL doxycycline, and kept on until DIV15, by doxycycline hemi-supplementation every 2 days. Finally, at DIV15, cells were fixed in 4% PFA for analysis. 

#### 5.3.2. Main Media Composition Was as Follows

Pro-proliferative medium: DMEM-F12 (Gibco), 1X Glutamax (Gibco), 1X N2 (Invitrogen), 1 mg/mL BSA, 0.6% glucose, 2μg/mL heparin (Stemcell technologies #7980), 20 ng/mL bFGF (Invitrogen #PHG0261), 20 ng/mL EGF (Invitrogen #PHG0311), 1XPen/Strept (Invitrogen #15140122), 10 pg/mL fungizone (Invitrogen #15290026)

Pro-differentiative medium: Neurobasal-A, 1XGlutamax (Gibco), 1XB27 supplement (Invitrogen), 25μM β-Mercaptoethanol, 2%Heat-Inactivated fetal bovine serum (FBS) (Euroclone), 1× Pen/Strept (Invitrogen #15140122), 10 pg/mL fungizone (Invitrogen #15290026).

### 5.4. Dendrite Morphometry

After image acquisition by an operator blind of sample identity and files randomization, neuronal silhouettes, limited to somas and dendrites, were generated with the Simple Neurite tracer plug-in [41], in ImageJ environment, by an operator blind of sample genotype. These silhouettes were analyzed by the Neurphology interactive plug-in [42], again in ImageJ environment. Three primary parameters were extracted: exN (exit-points number), enN (end-points number) and ∑l_i_ (total dendrite length). Then, as detailed in Figure 3, primary parameters were used for calculation of two derived indexes: anN (aligned nodes number), and l (internodal distance), in turn subject of subsequent statistical evaluation. Numerical calculations and statistical assessments were performed by Excel software.

### 5.5. Ca^2+^ Imaging Evaluation of Neuronal Activity

E16.5 neocortical neurons were obtained and cultured as described above. At DIV10, they were recorded at RT, under a Nikon Eclipse 80i microscope equipped with a 20× objective (NA = 0.35). Recordings were performed from 680 × 680 μm^2^ visual fields (binning 2), using an ORCA-Flash4.0 LT Digital CMOS (Hamamatsu #C11440) camera, managed by µManager Studio software (version: 2.0.0-gamma1 #20200529, [43]). GCaMP6s was used as a genetically encoded Ca^2+^ indicator, allowing the quantification of Ca^2+^ fluctuations within cell bodies. Fluorescence was excited at 488 nm with a mercury lamp. Excitation light was separated from the light emitted from the sample using a 395 nm dichroic mirror and ND filter (1/8). Images were acquired every 250 ms for 5–10 min with exposure time of 100 ms. Recorded images were analyzed off-line with Fiji (manually selecting ROIs around cell bodies) and Clampfit software (pClamp suite, 10.4.2 version, Molecular Devices LLC, US). Intracellular Ca^2+^ transients were expressed as fractional amplitude increases (ΔF/F_0_, where F_0_ is the baseline fluorescence level and ΔF is the rise over baseline). Fluorescence events whose amplitude exceed by >5 times noise standard deviation were taken into account. Inter-event interval (IEI) values were calculated computing the difference between consecutive event onset times. Three main parameters were evaluated: (a) average prevalence of active neurons; (b) median frequency of Ca^2+^ events (evaluation restricted to active neurons); and (c) cumulative distribution of inter- Ca^2+^ events-intervals (IEIs). Full details of statistical analysis of these parameters are provided in Legend to Figure 4A.

### 5.6. Extracellular Electrophysiological Recordings by MEAs

Commercial MEAs (Multichannel Systems GmBH, Reutlingen, Germany) were used to monitor the extracellular electrical activity in cortical neuronal cultures. Each MEA contains 120 titanium nitrate (TiN) microelectrodes with a diameter of 30 μm and an inter-electrode distance of 100 μm, arranged as a 12 × 10 regular layout. E16.5+DIV21 primary neuronal cultures on MEAs were set as described above. MEA electrodes detected extracellular action potentials from neurons located in their proximity. We employed an electronic multichannel amplifier (MEA2100-Mini-120-System, Multichannel Systems GmBH, Reutlingen, Germany) with 10–10,000 Hz bandwidth and an amplification factor of 1. Recordings of capped neuronal preparations were performed within a (dry) incubator, at 37 °C and 5% CO_2_ (C150, Binder GmbH, Tuttlingen, Germany). Extracellular raw electrical signals were sampled at 25 kHz/channel and digitized at 16 bits resolution by a MEA2100-Mini USB interface. The raw voltage traces were analyzed with custom scripts written in Julia as previously reported [44], to extract the time of occurrence of action potentials at each MEA microelectrode. For each recording channel a threshold for peak-detection was set, referring to background electrical noise [45]. Cumulative distributions of three parameters were evaluated: (a) interburst intervals (IBI), (b) burst durations and (c) spikes’ number per burst. Full details of statistical analysis of these parameters are provided in Legend to Figure 4B.

### 5.7. Quantitative RT-PCR

In each experimental session, aliquots of 6 × 10^5^ cells were processed for RNA extraction by Trizol^TM^ Reagent (ThermoFisher) according to manufacturer’s instructions. RNA preparations were treated by TURBO^TM^ DNase (2 U/μL) (Ambion^TM^) 1 h at 37 °C. At least 0.75 μg of genomic DNA-free total RNA from each sample was retro-transcribed by SuperScriptIII^TM^ (Invitrogen) in the presence of random hexamers, according to manufacturer’s instructions. 1/100 of the resulting cDNA was used as substrate of any subsequent qPCR reaction. Limited to intronless amplicons, negative control amplifications were run on RT(-) RNA preparations. PCR reactions were performed by SsoAdvanced SYBR Green Supermix^TM^ (Biorad), according to manufacturer’s instructions. The oligonucleotides employed in this study are listed in Appendix A. For each transcript under examination and each sample, cDNA was PCR-analyzed at least in technical triplicate and results averaged. When not otherwise specified, averages were further normalized against Gapdh. Experiments were performed at least in biological triplicates and analyzed by Student’s *t* test.

### 5.8. Immunofluorescent Assay

Neural cultures were fixed by 4% PFA for 15 min at RT, followed by three washes in 1X PBS. In all cases, samples were subsequently treated with blocking mix (1X PBS; 10% FBS; 1 mg/mL BSA; 0.1% Triton X100) for 1 h at RT. After that, incubation with primary antibody was performed in blocking mix, overnight at 4 °C. The day after, samples were washed in “1X PBS-0.1% Triton X-100” 3 times for 5 min and then incubated with a secondary antibody in blocking mix, for 2 h at RT. Samples were finally washed in 1X PBS for 5 min, 3 times and subsequently counterstained with DAPI (4′,6′-diamidino-2-phenylindole) and mounted in Vectashield Mounting Medium (Vector).

Primary antibodies employed in this study are listed in Appendix A. Immunoreactivity was revealed by Alexa Fluor 488 and 594-conjugated anti-mouse, -rat, -rabbit, chicken Abs (Invitrogen), used at 1:500.

### 5.9. Image Acquisition

Immunofluorescences were photographed on a Nikon TI-E apparatus, equipped with a Hamamatsu C4742-95 camera. 10× in air (Appendix A) and 20× in air (Appendix A, Appendix A, Appendix A, Appendix A and Appendix A) objectives were used. Pictures were acquired as .nd2 files by NIS software, they were further processed by Fiji software. For each independent biological replicate, at least 6 distinct fields were photographed by an operator blind of cells “genotype”. Images were analyzed after files randomization. In case of morphometric assays, each photographic field prevalently included one single, entire EGFP^+^ neuron. In case of clonal assays, single clones were identified, by an operator blind of Tubb3 and GFAP signal, as isolated groups of DAPI^+^ cells whose reciprocal distance is less than one cell diameter. Neuronal clones included only Tubb3^+^ cells, astroglial clones only Gfap^+^ cells, mixed clones include both Tubb3^+^ and Gfap^+^ cells. To build illustrative examples reported in Supplementary Figures, primary pictures were finally mounted by Photoshop software.

### 5.10. Statistical Analysis

Full details of data statistical analyses are provided in Legends to Figures.

## Figures and Tables

**Figure 1 ijms-23-01343-f001:**
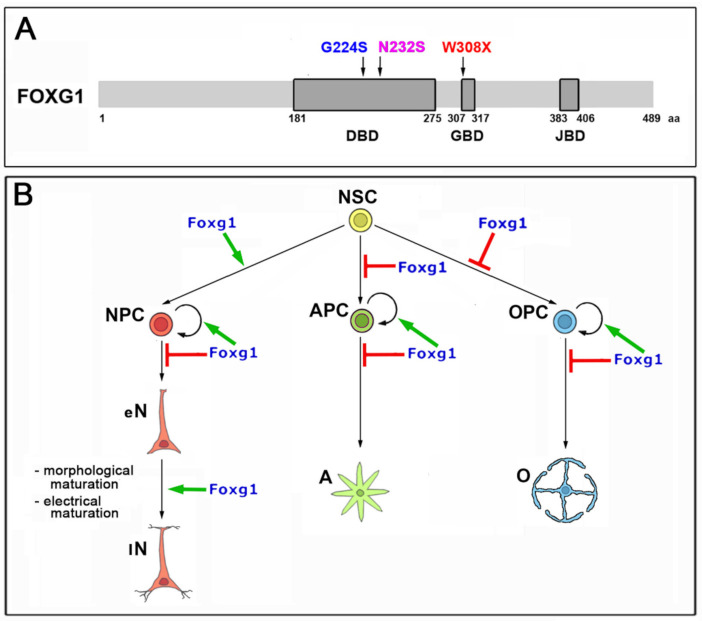
Foxg1: a structural and functional summary. (**A**) Foxg1 protein and selected variants of it. Boxed are the three Foxg1 domains, forkhead DNA binding domain (DBD), Groucho binding domain (GBD), and Jarid1B binding domain (JBD). On the top, the three structural mutations subject of this study. (**B**) Graphical summary of *Foxg1* impact on main subroutines of neocortical histogenesis. NSC, neural stem cells; NPC, neuron progenitor cells; eN, early neurons; lN, late neurons; APC, astrocyte progenitor cells; A, astrocytes; OPC, oligodendrocyte progenitor cells; O, oligodendrocytes.

**Figure 2 ijms-23-01343-f002:**
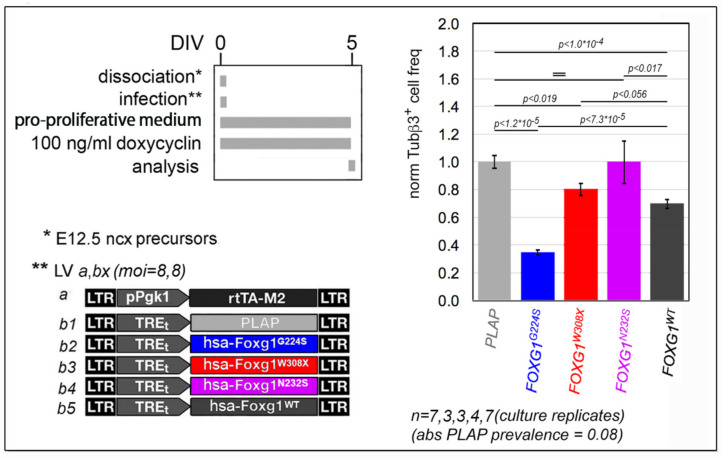
Impact of mutant *FOXG1* alleles on neuronogenic rates. To left, protocols and lentiviral vectors employed, to right, results. Shown are frequencies of Tubb3^+^ newborn neurons originating from neural precursors expressing *FOXG1^G224S^*, *FOXG1^N232S^*, *FOXG1^W308X^* and *FOXG1^WT^* alleles, in the presence of growth factors (GFs). Data normalized against *Plap*-expressing controls (control, absolute Tubb3^+^ cell frequency = 0.08). Statistical evaluation of results by *t*-test (one-tailed, unpaired). =, not significant. *n* is the number of biological replicates, i.e., independently cultured and engineered aliquots originating from a common neural pool.

**Figure 3 ijms-23-01343-f003:**
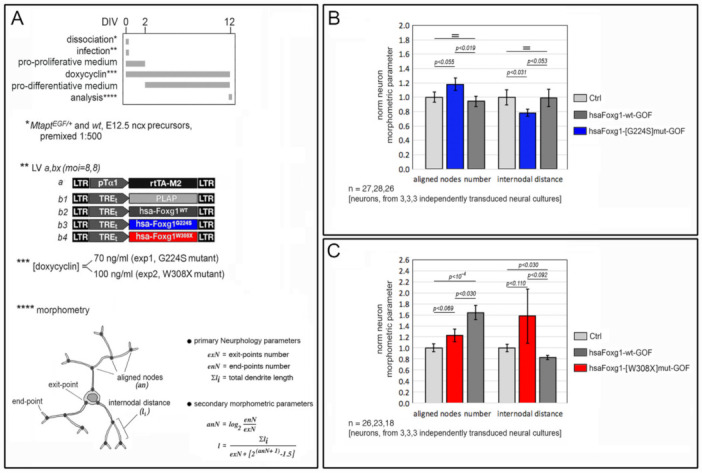
Impact of mutant *FOXG1* alleles on neuronal architecture. Protocols in (**A**), results in (**B**,**C**). Briefly, upon lentiviral engineering, *Mapt^EGFP/+^* neural cells were co-cultured with an excess of co-engineered/non-fluorescent neural cells, and then analyzed as follows: (1) *Mapt^EGFP/+^* neurons were profiled by αEGFP/αNF immunofluorescence; (2) pictures were skeletonized by Simple Neurite Tracer; (3) primary morphometric indices, *exN* (exit-points number), *enN* (end-points number) and ∑*l_i_* (total dendrite length), were extracted by neurphology; (4) secondary indices, *average anN* (aligned nodes number) and *average l* (internodal distance), were calculated by Excel, as shown. Finally, results were averaged and statistically evaluated. Shown are data normalized against *Plap*-expressing controls (absolute *anN_Plap_* = 1.91 ± 0.11 and 1.31 ± 0.08, in exp1 and exp2, respectively; absolute *l_Plap_* = (28.33 ± 2.95) μm and (30.11 ± 2.10) μm, in exp1 and exp2, respectively). Statistical evaluation of results by *t*-test (one-tailed, unpaired). =, not significant. *n* is the number of biological replicates, i.e., single neurons evenly taken from 3,3,3 independently cultured and engineered preparations, originating from a common neural pool.

**Figure 4 ijms-23-01343-f004:**
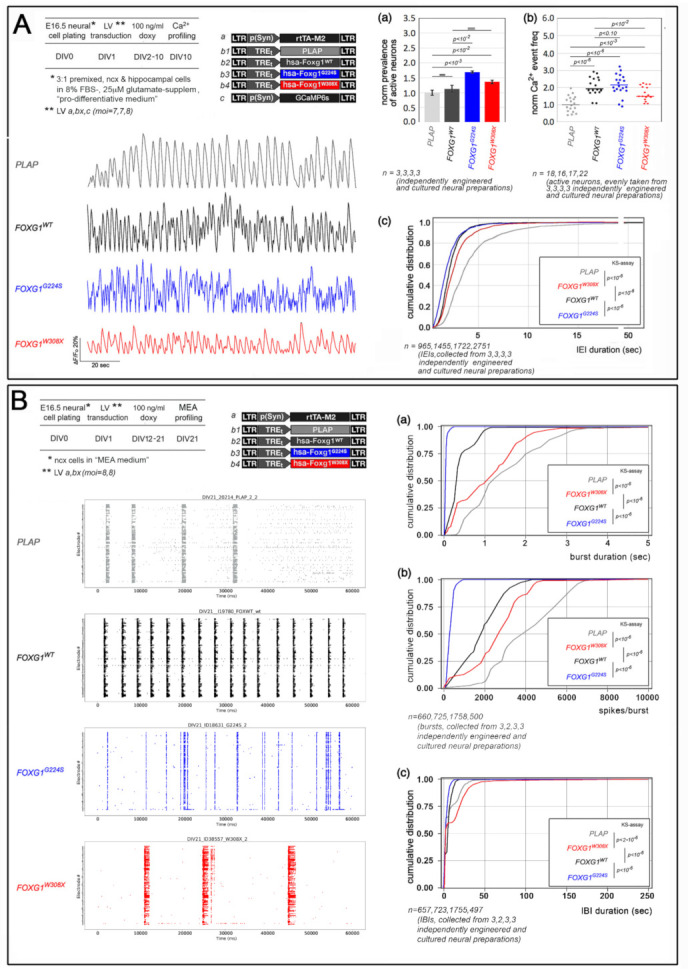
Impact of mutant *FOXG1* alleles on neuronal activity. (**A**) Functional profiling of E16.5+DIV10 neocortical neurons, overexpressing Tet^ON^-controlled, mutant *FOXG1* transgenes, by genetically encoded Ca^2+^ sensors. Top-left: protocols and lentiviral vectors employed; bottom-left: illustrative examples of neuronal Ca^2+^ traces; right: results. Shown are: (**a**) average prevalence of active neurons; (**b**) median frequency of Ca^2+^ events (evaluation restricted to active neurons); and (**c**) cumulative distribution of inter-Ca^2+^ events-intervals (IEIs). In (**a**) and (**b**), data normalized against *Plap*-expressing controls (absolute average prevalence of active neurons among *Plap*-expressing ones = 0.52; absolute median frequency of Ca^2+^ events in *Plap*-expressing neurons = 0.22 Hz). Statistical evaluation of results: in (**a**) and (**b**), by *t*-test (one-tailed, unpaired, by Excel software); in (**c**), by Kolmogorov–Smirnov test. *n* is the number of biological replicates, i.e.,: in (**a**), the number of independently cultured and engineered preparations originating from a common neural pool; in (**b**), the number of single neurons evenly taken from 3,3,3,3 independently cultured and engineered preparations, originating from a common neural pool; in (**c**), the cumulative number of IEIs collected from ΔF/F traces of such neurons. Scalebars in (**a**) represent s.e.m’s. (**B**) Functional profiling of E16.5+DIV21 neocortical neurons, overexpressing Tet^ON^-controlled, mutant *FOXG1* transgenes, by multi-electrode arrays (MEA). Top-left: protocols and lentiviral vectors employed; bottom-left: illustrative examples of MEA raster plots; right: results. Shown are cumulative distributions of: (**a**) burst durations (BDs), (**b**) number of spikes/burst (S/B), and (**c**) inter-burst interval (IBIs). Statistical evaluation of results by Kolmogorov–Smirnov test. *n* is the cumulative number of bursts (**a**,**b**) and inter-burst intervals (**c**) collected from 3,2,3,3 independently cultured and engineered preparations, originating from a common neural pool.

**Figure 5 ijms-23-01343-f005:**
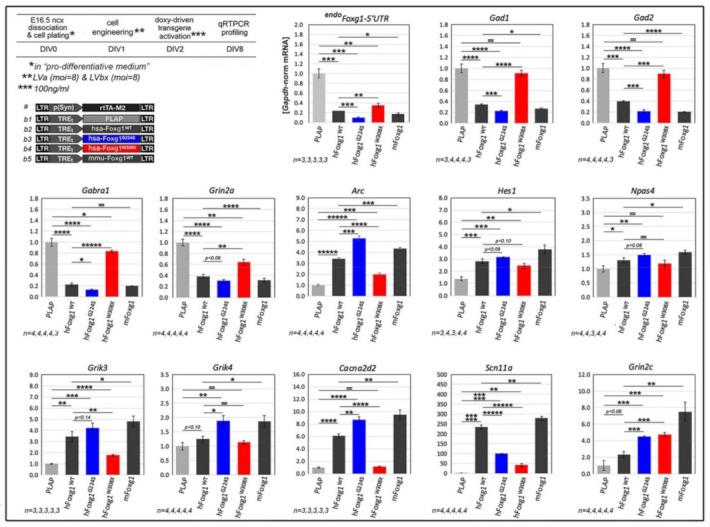
Impact of mutant *FOXG1* alleles on neuronal transcriptome, upon lentiviral delivery of transgenes encoding for them to *wild-type*, murine neuronal cultures. Top-left: protocols and lentiviral vectors employed; bottom-right: qRTPCR results. Data normalized against *Plap*-expressing controls. Statistical evaluation of results by *t*-test (one-tailed, unpaired). * *p* < 0.05, ** *p* < 0.01, *** *p* < 0.001, **** *p* < 10^−4^, ***** *p* < 10^−5^, ******* *p* < 10^−6^, = not significant. *n* is the number of biological replicates, i.e., the number of independently cultured and engineered preparations originating from a common neural pool. Scalebars represent s.e.m.

**Figure 6 ijms-23-01343-f006:**
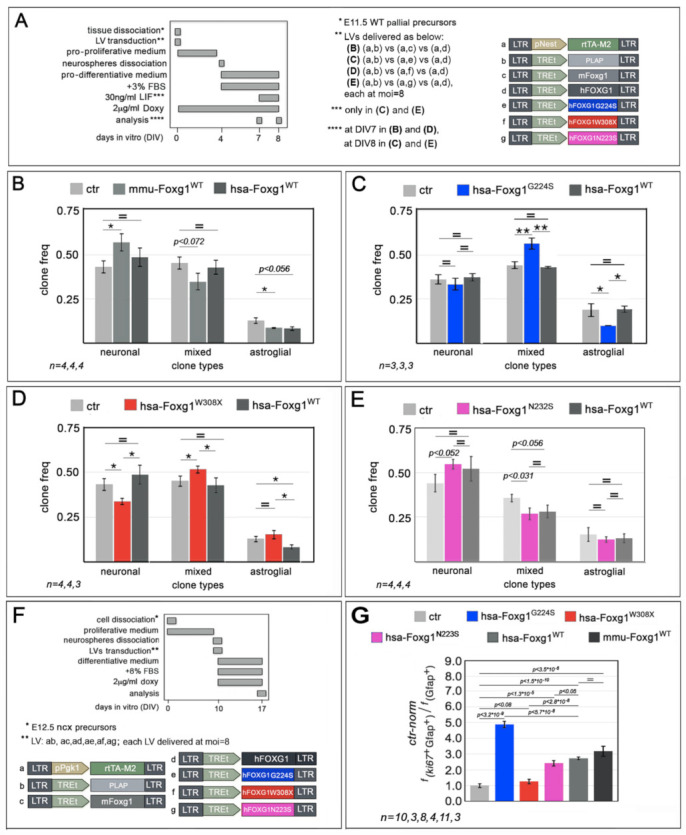
Impact of mutant *FOXG1* alleles on astrogenesis progression in murine, primary neocortical cultures. (**A**–**E**) Impact of different *FOXG1* alleles on fate choice, neuronal-vs-astroglial, by neocortical stem cells (NSCs), upon their lentiviral transduction with transgenes encoding for such alleles. In (**A**), protocols and lentiviral vectors employed, in (**B**–**E**), results. Shown are absolutes frequencies of neuronal, mixed and astroglial clones, generated by derivatives of E11.5 pallial NSCs upon acute genetic manipulation, intermediate expansion, and final differentiation at clonal densities. (**F**,**G**) Impact of different *FOXG1* alleles on mitogenic properties of astrocyte-committed progenitors, upon their lentiviral transduction with transgenes encoding for such alleles, as assessed by means of Ki67/Gfap-immunofluorescence. In (**F**), protocol and lentiviral vectors employed, in (**G**), results. Shown are frequencies of intermitotic Ki67^+^ cells among Gfap^+^ ones. Data normalized agaist *Plap*-expressing controls (absolute, control Ki67^+^Gfap^+^/Gfap^+^ ratio = 0.17). Throughout the Figure, statistical significance of results evaluated by *t*-test (1-tail, unpaired). * *p* < 0.05, ** *p* < 0.01, = not significant. *n* is the number of biological replicates, i.e aliquots of pre-pooled, independently lentivirus-transduced, and cultured neural cells. Scalebars represent s.e.m.

**Figure 7 ijms-23-01343-f007:**
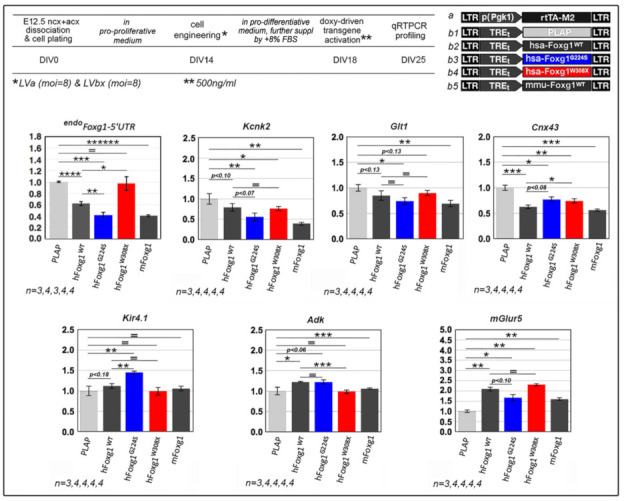
Impact of mutant *FOXG1* alleles on astroglial transcriptome, upon lentiviral delivery of transgenes encoding for them to *wild-type*, murine astroglial cultures. Top: protocols and lentiviral vectors employed; bottom: qRTPCR results. Data normalized against *Plap*-expressing controls. Statistical evaluation of results by *t*-test (one-tailed, unpaired). * *p* < 0.05, ** *p* < 0.01, *** *p* < 0.001, **** *p* < 10^−4^, ******* *p* < 10^−6^, = not significant. *n* is the number of biological replicates, i.e., the number of independently cultured and engineered preparations originating from a common neural pool. Scalebars represent s.e.m.

**Figure 8 ijms-23-01343-f008:**
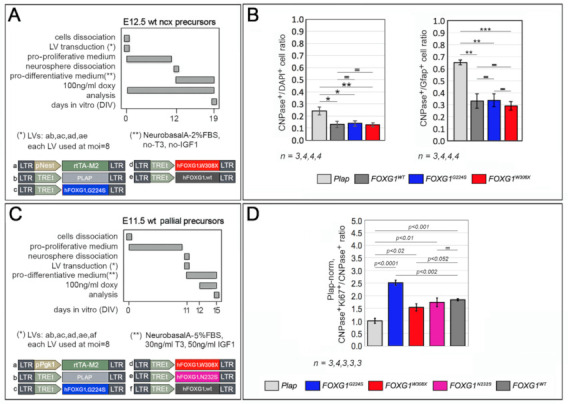
Impact of mutant *FOXG1* alleles on oligodendrogliogenesis. (**A**,**B**) Frequency of astroglial GFAP^+^ and oligodendroglial CNPase^+^ cells, and their ratio, among derivatives of early pallial precursors, acutely transduced with Tet^ON^-controlled lentiviral *FOXG1* transgenes, allowed to age in pro-proliferative conditions beyond the neuronogenic window, and finally allowed to differentiate. In (**A**) protocols and lentiviral vectors employed, in (**B**) results. (**C**,**D**) Impact of different *FOXG1* alleles on mitogenic properties of oligodendrocyte-committed progenitors, upon their lentiviral transduction with transgenes encoding for such alleles, as assessed by means of Ki67/CNPase-immunofluorescence. In (**C**), protocol and lentiviral vectors employed, in (**D**), results. Shown are frequencies of intermitotic Ki67^+^ cells among CNPase^+^ ones. Data normalized against *Plap*-expressing controls (absolute, control Ki67^+^CNPase^+^/CNPase^+^ ratio = 0.100). Throughout the Figure, statistical significance of results evaluated by *t*-test (1-tail, unpaired). * *p* < 0.05, ** *p* < 0.01, *** *p* < 0.001, =, not significant. *n* is the number of biological replicates, i.e., aliquots of pre-pooled, independently lentivirus-transduced and cultured neural cells. Scalebars represent s.e.m.

**Table 1 ijms-23-01343-t001:** Biological activities of *FOXG1* alleles: a synopsis.

Biological Process/Neurodevelopmental Parameter	*FOXG1^WT^*	*FOXG1^G224S^*	*FOXG1^W3028X^*	*FOXG1^N232S^*
	(vs-ctr)	(vs-*FOXG1^WT^*)	(vs-*FOXG1^WT^*)	(vs-*FOXG1^WT^*)
NSC commitment to neuronogenesis	↑	GOF	LOF	=
neuron birth rate	↓	GOF	LOF	LOF
dendritic elongation and arborization	↑	GOF	LOF	na
spontaneous neuronal activity	↑	GOF	LOF	na
modulation of select neuronal genes (1)	↓	GOF	LOF	na
modulation of select neuronal genes (2)	↑	GOF	LOF	na
NSC commitment to astrogliogenesis	↓	GOF	LOF	=
astroblast proliferation rate	↑	GOF	LOF	LOF
modulation of select astroglial genes (1)	↓	GOF	LOF	na
modulation of select astroglial genes (2)	↑	V	V	na
NSC progression to oligodendrocytes	↓	=	=	na
oligodendroblast proliferation rate	↑	GOF	LOF	=

na, not-assessed; GOF, gain-of-function; LOF, loss-of-function; V, variable; =, not-affected.

## Data Availability

Data supporting the findings of this study are available from the Authors upon reasonable request.

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
