# Peer review of "Multidimensional Functional Profiling of Human Neuropathogenic FOXG1 Alleles in Primary Cultures of Murine Pallial Precursors"

_ijms, 2022, doi:10.3390/ijms23031343_

Round 1

Reviewer 1 Report

Frisari et al. present here an extensive work regarding functional profiling of three FOXG1 alleles in vitro. The manuscript is very well written and the work greatly presented. I have just few suggestions for the authors.

  • Despite IJMS does not put limits on articles' length, this manuscript results very long and not very reader-friendly. The authors should consider to move part of the manuscript to the Supporting Information (SI). By the way, I suggest the authors to produce a proper SI file with all the figures and the related captions. 
  • In the Results section, the authors could make the text more readable by removing and distributing the statistical analysis in the caption of related figures. Also, "decrease by -XX%" is not correct, please remove the minus sign and correct it to "decrease by XX%"  
  • The choice of those three particular alleles is not very clear. Please, briefly add an explanation in the Introduction section.
  • Some abbreviations are not properly explicated, e.g. ASD-like, LOF, GOF, Plap.
  • Some references are not correctly reported in the text, especially in the Discussion and Conclusions section. Please, correct.
  • I would suggest the authors to split the Discussion and Conclusions  into two separated sections.
  • Lines 127- 141 are not justified neither present proper margins.

Finally, I would like to express my compliments to the authors for this very interesting work.

Reviewer 2 Report

The manuscript by Frisari et al., presents data on FOXG1 alleles in primary cultures. the experiments are well described and elegant. I have few minor comments:

  1. in the discussion it is stated "Remarkably, our study was performed in wild-type neocortical precursors, engineered to overexpress exogenous Foxg1 alleles and, therefore, characterized bycumulative Foxg1-mRNA levels well above the physiological baseline". the authors further discuss this point, but I was wondering if it could be possible to knockdown endogenous levels before lentiviral transfection, that would also better model heterozygous mutations?
  2. in the intriduction, please revise gene-protein nomenclature
  3. Figure 5: it is very difficult to read headings, please improve.
  4. in the abstract, at the end it is written :"namely a step
    propedeutic to timely delivery of precision treatments in utero". this sentence should be removed or modified. in utero treatments are far to be envisaged in this context and I agree that this manuscript increases the information towards personalized medicine, but not for "in utero treatments".
